# A Critical Appraisal of National and International Clinical Practice Guidelines Reporting Nutritional Recommendations for Age-Related Macular Degeneration: Are Recommendations Evidence-Based?

**DOI:** 10.3390/nu11040823

**Published:** 2019-04-11

**Authors:** John G. Lawrenson, Jennifer R. Evans, Laura E. Downie

**Affiliations:** 1Division of Optometry and Visual Science, City, University of London, Northampton Square, London EC1V OHB, UK; 2International Centre for Eye Health, Clinical Research Department, London School of Hygiene and Tropical Medicine, Keppel St, London WC1E 7HT, UK; Jennifer.Evans@lshtm.ac.uk; 3Department of Optometry and Vision Sciences, The University of Melbourne, Parkville, Victoria 3010, Australia; ldownie@unimelb.edu.au

**Keywords:** clinical practice guidelines, systematic reviews, age-related macular degeneration, nutritional supplements, diet, nutrition, AGREE II

## Abstract

Eye care professionals should have access to high quality clinical practice guidelines that ideally are underpinned by evidence from robust systematic reviews of relevant research. The aim of this study was to identify clinical guidelines with recommendations pertaining to dietary modification and/or nutritional supplementation for age-related macular degeneration (AMD), and to evaluate the overall quality of the guidelines using the Appraisal of Guidelines for Research and Evaluation II (AGREE II) instrument. We also mapped recommendations to existing systematic review evidence. A comprehensive search was undertaken using bibliographic databases and other electronic resources for eligible guidelines. Quality appraisal was undertaken to generate scores for each of the six AGREE II domains, and mapping of extracted nutritional recommendations was performed for systematic reviews published up to March 2017. We identified 13 national and international guidelines, developed or updated between 2004 and 2019. These varied substantially in quality. The lowest scoring AGREE II domains were for ‘Rigour of Development’, ‘Applicability’ (which measures implementation strategies to improve uptake of recommendations), and ‘Editorial Independence’. Only four guidelines used evidence from systematic reviews to support their nutritional recommendations. In conclusion, there is significant scope for improving current Clinical Practice Guidelines for AMD, and guideline developers should use evidence from existing high quality systematic reviews to inform clinical recommendations.

## 1. Introduction

Age-related macular degeneration (AMD) is an ocular disease affecting the central area of the retina (the macula), which is responsible for high-resolution daytime vision [1]. In the early stages of AMD, the macula shows characteristic sub-retinal lipoprotein deposits, known as drusen, and the monolayer of pigmented cells beneath the retina (retinal pigment epithelium) shows areas of hypo- and hyper-pigmentation. As the disease progresses, the retinal pigment epithelium can become atrophic, with secondary dysfunction and a loss of retinal photoreceptors [2]. Less commonly, new blood vessels grow beneath, or within, the retina; these vessels have a tendency to leak, causing disruption to the macular architecture and ultimately scar formation (neovascular AMD). Early AMD is typically asymptomatic; however, later stages of AMD can have a significant negative impact on visual function and quality of life [3]. AMD is currently the leading cause of severe vision impairment among people aged 50 and over in high-income countries [4]. With increasing life expectancy these numbers are projected to increase substantially in the future.

AMD is a complex disorder with several non-modifiable and modifiable risk factors [5]. The retina is particularly susceptible to oxidative stress, as a result of its high oxygen consumption and exposure to light; animal and cell culture studies have identified oxidative stress as a contributory factor for the development of AMD [6]. A large body of observational and experimental research in humans has investigated the association between dietary antioxidants and AMD, in particular whether an increased intake of antioxidant vitamin and minerals or specific carotenoids can prevent or slow the progression of the disease [7,8]. The role of particular essential fatty acids has also been investigated, including exploiting the anti-inflammatory properties of long-chain omega-3 fatty acids.

Based on accumulating research evidence, dietary modification and/or nutritional supplementation has been proposed as a simple and potentially cost-effective strategy for modifying the risk of AMD. Although there is a large body of research in this area, the quality of studies is highly variable and conflicting results have been reported. These factors pose challenges for clinicians, when attempting to provide evidence-based recommendations to patients about the relative risks and benefits of nutrition-based interventions. Clinical practice guidelines can be used to aid clinical decision-making in such circumstances. These documents should contain recommendations that are informed by a systematic review of the available research evidence, and a consideration of the benefits and harms of alternative interventions [9]. In the process of guideline development, developers either elect to perform their own systematic review of the literature, or to instead integrate the results from previously published reviews. Systematic reviews involve a process of systematically identifying, appraising and synthesising findings from all relevant research studies relating to a particular research question, with the intent of minimising the potential for bias. The objectives of the present study were to: 

(1) Identify clinical practice guidelines for AMD that contain nutritional recommendations, based upon a systematic literature search; 

(2) Evaluate the quality of these guidelines using the Appraisal of Guidelines for Research and Evaluation II (AGREE II) tool; and 

(3) Map clinical practice guideline recommendations, as related to nutrition and AMD, to relevant systematic reviews.

## 2. Materials and Methods

### 2.1. Eligibility Criteria

We included national and international clinical practice guidelines for AMD that contained nutritional recommendations, including dietary modification or nutritional supplementation, as strategies to prevent or slow the progression of the disease. 

### 2.2. Search Strategy

We searched the Ovid MEDLINE and Embase bibliographic databases from January 1999 to January 2019 using search filters for identifying clinical guidelines that were developed by the Canadian Agency for Drugs and Technologies in Health (https://www.cadth.ca) (see Appendix A). In addition to the bibliographic database searches, we also searched Guideline Central Summaries (https://www.guidelinecentral.com/summaries/) and the Turning Research into Practice (TRIP) database (https://www.tripdatabase.com/), and undertook a search of the webpages of national and international professional organisations for ophthalmology and optometry. The search was not limited by language and we used Google Translate to extract recommendations and appraise guidelines that were not written in English.

### 2.3. Study Selection and Data Extraction

Following removal of duplicates, two reviewers (JL/LD) independently screened the titles and abstracts identified from the bibliographic searches and resolved any discrepancies by discussion and consensus. We obtained full-text copies of potentially eligible guidelines and these were independently assessed by all three authors, working in pairs, to determine whether they met the inclusion criteria. Reasons for exclusion were documented at this stage. Two reviewers (all three authors working in pairs) independently extracted general characteristics of each guideline (e.g., author/organisation, country, year of publication, target audience) and details of the specific nutritional recommendations.

### 2.4. Appraisal of Clinical Guidelines

The quality of each clinical guideline was independently evaluated by two reviewers (all three authors working in pairs) using the Appraisal of Guidelines for Research and Evaluation (AGREE) II tool (https://www.agreetrust.org/). The AGREE II instrument evaluates the process of practice guideline development, and the quality of reporting, using 21 items organised into six key domains, as follows: (1) Scope and Purpose; (2) Stakeholder Involvement; (3) Rigour of Development; (4) Clarity of Presentation; (5) Applicability; and (6) Editorial Independence. Table 1 provides further details about the content of these domains. Each item within the AGREE II tool comprises a quality statement/concept, which is scored using a 7-point Likert rating scale. A score of ‘1’ was given when there was no information that was relevant to the item or where the concept was very poorly reported. A score of ‘7’ was given if the quality of reporting met the full criteria, as defined in the AGREE II User’s Manual. The overall scores for each of the six domains were calculated by summing up all the scores of the individual items for that domain, and then scaling the total as a percentage of the maximum possible score for each domain. Therefore higher scores indicate higher guideline quality. An intra-class correlation coefficient (ICC) was used to evaluate overall inter-rater agreement.

### 2.5. Mapping Clinical Guideline Recommendations to Systematic Review Evidence

Given the predominance of systematic reviews in the hierarchy of clinical evidence [10], we undertook an exercise to map the extracted guideline recommendations relating to diet or nutritional supplementation, in the context of AMD, to the evidence derived from systematic reviews. We also assessed whether recommendations were linked to particular citations (e.g., systematic reviews, individual randomised controlled trials (RCTs), or other study designs). To identify relevant systematic reviews on nutritional interventions for AMD we used two recently published studies [11,12] that identified and critically appraised systematic reviews of AMD interventions. Lindsley et al. [11] identified 47 systematic reviews published between 2001 and 2014, of which 9 (19%) evaluated dietary supplements. A more recent study by Downie and colleagues [12] found 71 systematic reviews published between 2003 and 2017, with 10 (12%) relating to nutritional interventions in AMD. These studies identified 11 unique reviews [13,14,15,16,17,18,19,20,21,22,23], of which over 50% were published after 2014.

The process of mapping systematic reviews to guideline recommendations relating to nutrition was performed by a single reviewer and then independently checked by a second reviewer.

## 3. Results

### 3.1. Search Results

The searches identified 868 potentially relevant records. Following title and abstract screening, 838 were excluded. After a full-text review of 30 potentially eligible records, 17 were excluded as they did not contain recommendations for nutritional interventions. Thirteen guidelines were included in the final analysis. Figure 1 shows the flow diagram for guideline selection. 

### 3.2. Characteristics of Included Guidelines and Nutritional Recommendations

Table 2 summarises the characteristics of 13 national guidelines from the United Kingdom [24,25], United States [26,27,28], Canada [29,30], Australia [31], New Zealand [32], Spain [33], Germany [34], and the Philippines [35], and an international ophthalmology guideline [36], that were published between 2004 and 2019. Three guidelines were produced by government organisations [24,25,32], while the remainder were developed by professional societies or associations for specific eye care professions.

Eight guidelines [24,26,27,28,29,31,34,35] provided nutritional recommendations (either positive or negative) for reducing the risk of developing AMD (primary prevention) (Table 3). Guidelines from Canada [29], Germany [34], and Australia [31] included recommendations on the value of dietary modification for primary prevention. With regard to supplements, a guideline produced by an expert panel of optometrists in the United States [28] recommended the use of xanthophil supplements (lutein, zeaxanthin, and mesozeaxanthin) in ‘sub-clinical’ AMD and another suggested that supplementation with antioxidants may be beneficial for those who are nutritionally deficient [26]. Five guidelines specifically stated that there was evidence that use of high- dose anti-oxidant vitamin and mineral supplements was not beneficial for primary prevention (Table 3) [24,27,31,34,35].

All of the included guidelines provided recommendations regarding nutritional strategies for secondary prevention (i.e., slowing the progression of AMD) (Table 4). Five guidelines [24,28,30,31,33] included dietary advice, consisting of encouraging people with AMD to eat a healthy diet and, in particular, to increase consumption of foods rich in the macular carotenoids (lutein and zeaxanthin) and/or eat more oily fish as a source of omega-3 essential fatty acids. With regard to nutritional supplements, all guidelines addressed the topic of antioxidant vitamin and mineral supplements; the vast majority referred to supplements containing the combinations of antioxidant vitamins, zinc and carotenoids used in the Age-Related Eye Disease Studies (AREDS) or AREDS2 studies [7,8]. Only the U.K. National Institute for Health and Care Excellence (NICE) guideline [25] did not advocate the use high-dose antioxidant vitamin and mineral supplements for the secondary prevention of AMD. The NICE guideline committee concluded that “the current clinical evidence was not able to demonstrate a clear treatment benefit of antioxidant vitamin and mineral supplement for people with early AMD and was insufficient to make a strong recommendation on the use of these supplements”. The majority of guidelines acknowledged the risks as well as the benefits of antioxidant vitamin and mineral supplementation. Although acknowledging the lack of evidence, two guidelines [28,34] stated that omega-3 fatty acid supplements could be considered. By contrast, the NICE committee [25] concluded that “omega-3 fatty acid supplementation had no meaningful effect on AMD progression and visual acuity, and that therefore no recommendations could be made on this topic”.

### 3.3. Clinical Guideline Quality Scores

The overall inter-rater agreement on scoring of the individual items was excellent (correlation coefficient: 0.81 (95% confidence interval (CI) 0.76 to 0.84)).

The quality of each guideline, across each of the AGREE II domains, is shown in Table 5. The median percentage scores (%) for different domains ranged from 13% to 75%, with substantial heterogeneity in quality between guidelines. Only two of six domains had a median score over 50% (‘Scope and Purpose’ and ‘Clarity of presentation’). The lowest scoring domains were ‘Applicability’ (median 13% (range 0 to 98%)), ‘Editorial Dependence’ (median 20.8% (range 0 to 100%)), and Rigour of Development (median 20.4% (range 9.2 to 95.9%)). For the ‘Rigour of Development’ domain, only four of the 13 guidelines achieved a score ≥50%; this was largely due to a failure to demonstrate that systematic methods had been used to search for relevant evidence, no description of the criteria used for selecting and/or evaluating the evidence, and no information on the methods used by the developers to formulate recommendations.

### 3.4. Mapping Clinical Guideline Recommendations to Systematic Review Evidence

We extracted individual recommendations relating to nutritional strategies for both preventing and slowing the progression of AMD (see Table 3 and Table 4 for an overview). For some guidelines, no evidence was provided to support nutritional recommendations [33,36]. For other guidelines, individual studies (i.e., RCTs or prospective cohort studies) were cited. Only four guidelines [24,25,30,34] included evidence from systematic reviews despite the availability of reliable systematic reviews for all of the extracted nutritional recommendations [13,14,15,16,17,18,19,20,21,22,23]. 

## 4. Discussion

Nutritional supplements for ‘eye health’ are marketed at the general population, and are also widely recommended by optometrists and ophthalmologists for people who have clinical signs of AMD [37,38]. Clinical practice guidelines are a useful method for presenting evidence-based recommendations to health care professionals, and are intended to be used to inform clinical decision-making and reduce potential variations in practice. The aims of the current study were to identify all clinical guidelines for AMD that include recommendations relating to diet and/or nutrition; to evaluate the methodological quality of these guidelines using the AGREE II tool (the most widely used instrument for appraising clinical practice guidelines); and to investigate whether specific nutritional recommendations were underpinned by evidence from systematic reviews.

We identified 13 clinical practice guidelines meeting our inclusion criteria. The overall quality of these guidelines was judged to be low to moderate, with the median percentage scores for four of the six AGREE II quality domains being below 50%. The lowest scoring domains were ‘Rigour of Development’, ‘Applicability’, and ‘Editorial Independence’. Low scores in these domains are of major concern, as they relate to: the process by which relevant research evidence is gathered and synthesised; the methods used to formulate the recommendations; the likely barriers and facilitators to implementation of the guideline recommendations; and how conflicts of interest are managed. It is important that guideline users are able to identify the evidence underpinning each recommendation, however many of the guidelines failed to make an explicit link to the evidence used to formulate recommendations. Transparency with respect to the reporting of conflicts of interest amongst guideline development panels is also essential, to avoid the potential for biased recommendations. Patient engagement in the development of the clinical practice guidelines was also generally poor, despite standards (including those defined by the World Health Organisation) [39] recommending the inclusion of patients on guideline development panels. The intent of this engagement is to ensure a focus on patient-centred guidelines that can enhance the quality of care [40].

Only one guideline (the National Institute for Health and Care Excellence (NICE) guideline NG82) achieved high scores in all of the AGREE II assessment domains. This is perhaps unsurprising, since the NICE guideline development manual states that NICE guidelines “are based on internationally accepted criteria of quality as detailed in the Appraisal of Guidelines of Research and Evaluation II (AGREE II) instrument”. The development and implementation of high quality clinical practice guidelines requires substantial time, expertise and resources. Less formal guideline development groups, such as those produced by ‘expert panels’ or professional organisations, may lack the resources and/or methodological expertise to produce guidelines of the highest quality. In such circumstances, adapting existing high quality guidelines to local contexts may be an alternative to de novo development. The AMD Preferred Practice Patterns (PPP) Philippines (2016) [35] is an example of where this strategy has been adopted. This guideline was prepared by a panel from the Vitreo-Retina Society of the Philippines for the Philippine Academy of Ophthalmology, by adapting recommendations from the American Academy of Ophthalmology (AAO) PPP for AMD (updated in 2015).

Many robust systematic reviews have considered the merit of a variety of dietary and nutritional recommendations for AMD. In the present study, we used the systematic evidence searches performed by Lindsley et al. [11] and Downie et al. [12] to identify relevant systematic reviews, up to March 2017. These studies identified 11 unique systematic reviews on nutritional interventions for AMD, published between 2007 and 2016. Four of these reviews were published in the Cochrane Library. Mapping of clinical practice guideline recommendations to systematic review evidence showed that four of the included guidelines [24,25,30,34] made reference to systematic reviews to support their nutritional recommendations. Other guidelines included evidence from individual RCTs or non-randomised trials only. In some cases, recommendations were provided with no supporting citations. Although it is likely that a proportion of the systematic review evidence would not have been available to the panels developing the three earliest guidelines, published between 2004-2009 [26,33,36], this evidence would certainly have been available to later panels and could have been used to inform their recommendations. Inconsistent uptake of evidence from systematic reviews by decision-makers and guideline developers is known to be related to several factors, including lack of awareness, lack of access and lack of familiarity [41]. The outcome of failing to use the best-available research evidence (i.e., systematic reviews) to inform practice guidelines is that this time-intensive, rigorous research effort is wasted. Furthermore, the entire rationale for the guideline to support evidence-based practice, and optimise patient outcomes is potentially not achieved.

Of the nine guidelines that considered nutritional approaches for the primary prevention of AMD, most considered the potential merit of dietary modification, although some also made recommendations in relation to nutritional supplementation. Several epidemiological studies have investigated whether specific dietary patterns and/or foods are associated with a reduced risk of developing AMD. A meta-analysis of prospective cohort, case-control and cross-sectional studies concluded that consumption of two or more servings of oily fish per week was beneficial in the primary prevention of AMD [13]. Another recent systematic review reported that a high consumption of vegetables rich in carotenoids and oily fish containing omega-3 fatty acids was beneficial for those at risk of AMD [42]. However, emphasising the potential difference in nutritional benefit(s) derived from whole foods versus supplementation, consuming anti-oxidant supplements does not prevent the primary onset of AMD [16]. It is therefore concerning that a recent (2017) U.S. ‘guideline’ [28], produced by a panel of optometrists, recommended prescribing xanthophil supplements (i.e., lutein, zeaxanthin and mesozeaxanthin) to patients with ‘sub-clinical’ AMD, as “it is better to prescribe a supplement than not to prescribe a supplement”. Although termed a ‘guideline’, this document achieved the poorest quality scoring of all those included, with characteristics of a ‘clinical viewpoint’ rather than a ‘guideline’ per se.

Nutritional strategies for secondary prevention of AMD (i.e., slowing progression of the disease) were included in all of the clinical guidelines. Five guidelines [24,28,33,34] included dietary advice, consisting of recommendations for people with AMD to eat a healthy balanced diet and, specifically to increase the consumption of foods rich in the macular carotenoids and/or eat more fish (as a source of omega-3 fatty acids). A Mediterranean diet has been linked to a reduced risk of AMD progression [43]. Epidemiological studies suggest that a high dietary intake of omega-3 fatty acids is associated with a significant reduction in the risk of both intermediate [44,45] and late-stage AMD [46,47]. However, the best-available research evidence does not support long-chain omega-3 fatty acid supplementation for slowing disease progression [15]. Despite this, and acknowledging that there is a lack of evidence to substantiate such a position, two recent clinical guidelines [28,34] included a recommendation for clinicians to consider this approach.

With respect to high-dose anti-oxidant vitamin and mineral supplements for managing AMD, there were divergent recommendations. Notably, only the U.K. NICE guideline [25] did not advocate prescribing supplements containing the formulation of antioxidant vitamins, zinc and carotenoids investigated in the Age-Related Eye Disease Studies (AREDS [7] and AREDS2 [8]). This recommendation was based upon the committee’s assessment of the limitations of the current, best-available evidence, and a need for “further research in this area”. Indeed, there remain several key unanswered questions. For example, the minimum effective dose required for a given antioxidant to impart a potential retinoprotective effect remains unclear. Whether a single component, or a combination of components, represents the optimal formulation is also uncertain. The NICE guideline committee recommended “a well conducted randomised trial […] to provide an evidence base for the benefits and risks of individual components of the antioxidant supplements, and provide the ability to establish the treatment effect of antioxidant supplementation (the AREDS2 formula) on AMD progression by comparing AREDS2 formula with no treatment (for instance normal diet)”. Although many of the guidelines considered the potential contraindications and/or side effects of high-dose antioxidant vitamin and mineral supplements, four of the guidelines did not articulate these risks [28,31,32,36]. Given that the decision to prescribe an intervention should balance the potential risks versus benefits of treatment, lack of this key information to guide clinical decision-making is a major shortcoming.

A key strength of the current study was the comprehensive search to identify eligible clinical practice guidelines. Data extraction and quality appraisal were conducted independently by two reviewers, using a recognised assessment tool (AGREE II), which has established metrics of reliability and validity [48]. Overall, the reporting of methodological details for clinical guideline development was poor and therefore it was not possible to make a judgement as to whether the guidelines had used an appropriate evidence-based approach in their development.

## 5. Conclusions

Despite the availability of robust systematic reviews evaluating the efficacy and safety of nutritional interventions for AMD, this study found evidence that these resources are infrequently used to support recommendations in AMD clinical practice guidelines. Consequently, guidelines often present conflicting recommendations that could lead to variations in clinical care. The AGREE II quality evaluations of the included guidelines identified several key areas that require improvement, particularly the rigour of development, managing potential conflicts of interests, and presenting strategies for implementing guideline recommendations into daily practice.

## Figures and Tables

**Figure 1 nutrients-11-00823-f001:**
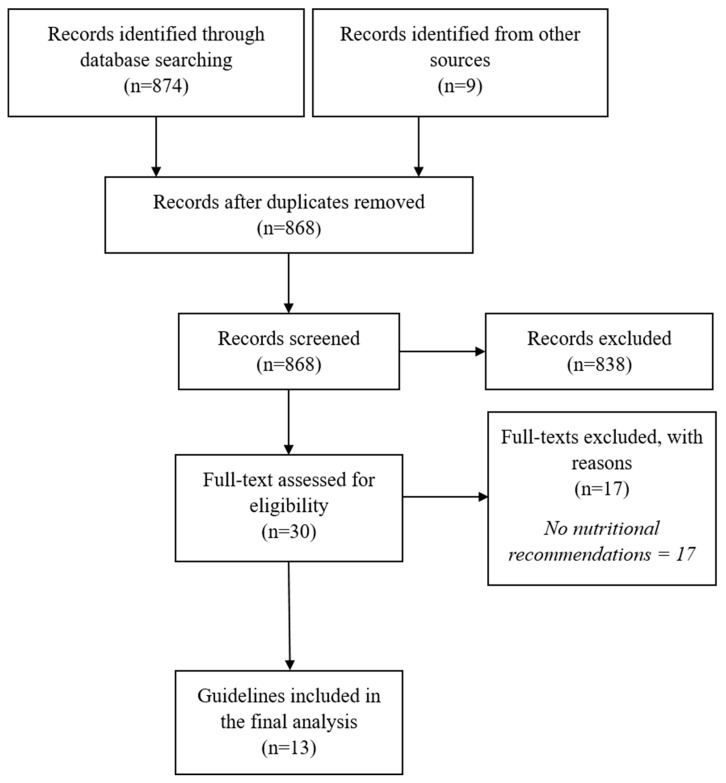
Selection process for identifying relevant clinical practice guidelines.

**Table 1 nutrients-11-00823-t001:** Clinical guideline quality domains used in the Appraisal of Guidelines for Research and Evaluation II (AGREE II) tool.

Domain	Content
Scope and Purpose (3 items)	Concerned with the overall aim of the guideline, the specific health questions, and the target population
Stakeholder Involvement (3 items)	Focuses on the extent to which the guideline was developed by the appropriate stakeholders and represents the views of its intended users
Rigour of Development (8 items)	Relates to the process used to gather and synthesize the evidence, the methods to formulate the recommendations, and to update them
Clarity of Presentation (3 items)	Deals with the language, structure, and format of the guideline
Applicability (4 items)	Pertains to the likely barriers and facilitators to implementation, strategies to improve uptake, and resource implications of applying the guideline
Editorial Independence (2 items)	Concerned with the formulation of recommendations not being unduly biased with competing interests

**Table 2 nutrients-11-00823-t002:** General characteristics of the included age-related macular degeneration (AMD) clinical practice guidelines.

Organisation	Reference	Year	Country	Target Audience
American Optical Association	[26]	2004	United States	Optometrists
International Council of Ophthalmology	[36]	2007	International	Optometrists
Spanish Retina and Vitreous Society (SERV)	[33]	2009	Spain	Ophthalmologists
Canadian Expert Consensus	[30]	2012	Canada	Ophthalmologists
German Ophthalmological Society (in German)	[34]	2014	Germany	Ophthalmologists
Eye Health Council of Ontario	[29]	2015	Canada	Health and eye care professionals
National Health Committee	[32]	2015	New Zealand	Healthcare professionals involved in the diagnosis and management of AMD
American Academy of Ophthalmology	[27]	2015	United States	Ophthalmologists
Vitreo-retina Society of the Philippines (VRSP)	[35]	2016	Philippines	Ophthalmologists
National Institute for Health and Care Excellence (NICE; Clinical Knowledge Summaries)	[24]	2016	United Kingdom	Primary care healthcare professionals
Clinical Advisory Committee [28]	[28]	2017	United States	Optometrists
NICE (NG82)	[25]	2018	United Kingdom	Healthcare professionals involved in the diagnosis and management of AMD
Optometry Australia	[31]	2019	Australia	Optometrists

**Table 3 nutrients-11-00823-t003:** Nutritional recommendations for primary prevention of age-related macular degeneration (AMD).

Clinical Guideline	Dietary Advice	Use of Antioxidant or Mineral Supplements	Use of Omega-3 Fatty Acid Supplements	Contraindications or Side Effects of Supplements	Systematic Review Cited with Recommendation
American Optical Association 2004	NR	✓ ^4^	NR	✓	None
International Council of Ophthalmology 2007	NR	NR	NR	N/A	N/A
Spanish Retina and Vitreous Society 2009	NR	NR	NR	N/A	N/A
Canadian Expert Consensus 2012	NR	NR	NR	N/A	N/A
German Ophthalmological Society 2014	✓ ^1^	✓ ^5^	NR	N/A	Yes
Eye Health Council of Ontario (Canada) 2015	✓ ^2^	✓ ^5^	NR	N/A	None
National Health Committee (New Zealand) 2015	NR	NR	NR	N/A	N/A
American Academy of Ophthalmology 2015	NR	✓ ^5^	NR	N/A	None
Vitreo-Retinal Society of the Philippines 2016	NR	✓ ^5^	NR	N/A	None
NICE Clinical Knowledge Summary (CKS) 2016	NR	✓ ^5^	NR	N/A	Yes
Clinical Advisory Committee (United States) 2017	NR	✓ ^6^	NR	NR	None
NICE Guideline (NG82) 2018	NR	NR	NR	N/A	N/A
Optometry Australia 2019	✓ ^3^	✓ ^7^	NR	NR	None

Abbreviations: ✓, recommendation included; NICE, National Institute for Health and Care Excellence; NR, no recommendation. Guideline recommendations: ^1^ Balanced diet; ^2^ A diet high in green, leafy vegetables (rich in antioxidants and carotenoids); ^3^ A diet high in macular carotenoids (zeaxanthin and lutein) and omega-3 fatty acids; ^4^ Antioxidant nutrient supplements (particularly for nutritionally deficient); ^5^ Evidence of no benefit for antioxidant vitamin and/or mineral supplements for primary prevention; ^6^ Supplement containing xanthophylls (lutein, zeaxanthin, meso-zeaxanthin) for ‘sub-clinical’ AMD. ^7^ Supplements not currently recommended for people with normal ageing changes.

**Table 4 nutrients-11-00823-t004:** Nutritional recommendations for secondary prevention (i.e., slowing the progression) of age-related macular degeneration (AMD).

Clinical Guideline	Dietary Advice	Use of Antioxidant or Mineral Supplements	Use of Omega-3 Fatty Acid Supplements	Contraindications or Side Effects of Supplements	Systematic Review Cited with a Recommendation
American Optical Association 2004	NR	✓ ^6^	NR	✓	None
International Council of Ophthalmology 2007	NR	✓ ^7a^	NR	NR	None
Spanish Retina and Vitreous Society 2009	✓ ^1^	✓ ^7a^	NR	✓	None
Canadian Expert Consensus 2012	✓ ^2^	✓ ^7a^	NR	✓	Yes
German Ophthalmological Society 2014	NR	✓ ^7a^	✓ ^10^	✓	Yes
Eye Health Council of Ontario (Canada) 2015	✓ ^3^	✓ ^7a^	✓ ^10^	✓	None
National Health Committee (New Zealand) 2015	NR	✓ ^7a^	NR	NR	None
American Academy of Ophthalmology 2015	NR	✓ ^7a^	NR	✓	None
Vitreo-Retina Society of the Philippines 2016	NR	✓ ^7a^	NR	✓	None
NICE Clinical Knowledge Summary (CKS) 2016	✓ ^3^	✓ ^7a^	NR	✓	Yes
Clinical Advisory Committee (United States) 2017	✓ ^4^	✓ ^7a^	✓ ^9^	NR	None
NICE Guideline (NG82) 2018	NR	✓ ^8^	✓ ^10^	✓	Yes
Optometry Australia 2019	✓ ^5^	✓ ^7b^	NR	NR	None

Abbreviations: ✓, recommendation included; NICE, National Institute for Health and Care Excellence; NR, no recommendation. Guideline recommendations: ^1^ Diet rich in carotenoids (lutein and zeaxanthin) and omega-3 fatty acids; ^2^ Dietary intake of antioxidants, docosahexaenoic acid, and omega-3 fatty acids; ^3^ Diet high in fresh fruit and green, leafy vegetables (rich in antioxidants and carotenoids); ^4^ Consume oily fish rich in DHA and follow healthier eating styles e.g., Mediterranean diet; ^5^ A diet rich in green leafy vegetables, fish and antioxidants should be encouraged; ^6^ Antioxidant nutrient supplements; ^7a^ Age-Related Eye Disease Studies (AREDS) or AREDS2 supplement recommended; ^7b^ AREDS or AREDS2 supplement may be beneficial, and should be discussed in conjunction with the patient’s general medical practitioner. ^8^ No evidence that benefit of AREDS supplement outweighs the risk; ^9^ Omega-3 fatty acid supplement recommended; ^10^ No evidence for benefit of omega-3 fatty acid supplements.

**Table 5 nutrients-11-00823-t005:** Quality, measured in %, of the included AMD clinical guidelines, based on the six assessment domains of the Appraisal of Guidelines for Research and Evaluation II (AGREE II) appraisal tool. The scores for each of the six domains were calculated by summing up all the scores of the individual items for that domain, and then scaling the total as a percentage of the maximum possible score for each domain.

Clinical Guideline	AGREE II Domains (%)					
	Scope and Purpose	Stakeholder Involvement	Rigour of Development	Clarity of Presentation	Applicability	Editorial Independence
American Optical Association 2004	52.8	30.6	9.2	30.6	10.9	87.5
International Council of Ophthalmology 2007	58.3	30.6	10.2	58.3	2.2	0.0
Spanish Retina and Vitreous Society 2009	63.9	41.7	21.4	36.1	13.0	0.0
Canadian Expert Consensus 2012	75.0	47.2	36.7	75.0	28.3	20.8
German Ophthalmological Society 2014	30.6	25.0	19.4	52.8	0.0	0.0
Eye Health Council of Ontario (Canada) 2015	61.1	22.2	14.3	41.7	13.0	20.8
National Health Committee (New Zealand) 2015	75.0	13.9	20.4	55.6	30.4	0.0
American Academy of Ophthalmology 2015	83.3	63.9	72.4	94.4	17.4	75.0
Vitreo-Retina Society of the Philippines 2016	83.3	44.4	56.1	88.9	13.0	45.8
NICE Clinical Knowledge Summary (CKS) 2016	88.9	83.3	68.4	91.7	39.1	87.5
Clinical Advisory Committee (United States) 2017	52.8	22.2	9.2	44.4	4.3	0.0
NICE Guideline (NG82) 2018	94.4	88.9	95.9	100.0	97.8	100.0
Optometry Australia 2019	83.3	41.7	19.4	75.0	19.6	0.0
Median (range)	75.0 (range 30.6 to 94.4%)	41.7 (range 13.9 to 88.9%)	20.4 (range 9.2 to 95.9)	58.3 (range 30.6 to 100%)	13.0 (range 2.2 to 97.8%)	20.8 (range 0 to 100%)

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
