# Peer review of "A Critical Appraisal of National and International Clinical Practice Guidelines Reporting Nutritional Recommendations for Age-Related Macular Degeneration: Are Recommendations Evidence-Based?"

_nutrients, 2019, doi:10.3390/nu11040823_

Round 1
Reviewer 1 Report
A critical appraisal of national and international clinical practice guidelines reporting nutritional recommendations for age-related macular degeneration: are recommendations evidence based'. It gives a good overview of clinical guidelines and explores their overall quality. Furthermore, it is well written and easy to follow.
I only have these three comments:
Paragraph 3.1 - The number of records mentioned are not the same as in Figure 1.
Please update the in text references so that they align with the journal's guidelines. Sometimes there is a space before the reference, some references are not merged etc.
A fullstop is missing at the end of the first paragraph on page 11.
Author Response
1. Paragraph 3.1 - The number of records mentioned are not the same as in Figure 1.
We have amended the paragraph describing the search results so that the numbers match those in Figure 1.
2. Please update the in text references so that they align with the journal's guidelines. Sometimes there is a space before the reference, some references are not merged etc.
The spacing is now consistent in the revised manuscript and the references have been merged as suggested
3. A full stop is missing at the end of the first paragraph on page 11.
A full stop has been added at the end of the paragraph
Reviewer 2 Report
The authors evaluated 13 guidelines concerning nutrition of AMD using AGREEII, and concluded that there have been no reliable guidelines for clinicians to educate patients on nutrition in daily practice.
I feel that this is a very interesting manuscript. The authors showed that some guidelines had low score for Rigor Development. This is a fundamental problem as a guideline. I hope more reliable guidelines based on the evidence will be established in a near future. The prevention of AMD is difficult, because AMD is a multifactorial disease. More investigation on the effect of nutrients for the primary and secondary prevention of AMD would be needed.
Minor criticisms;
The first two paragraphs in Discussion overlapped to Introduction. I think the appropriate brief summary of the present results is preferable for readers at the beginning of Discussion.
I found no other criticisms.
Author Response
The first two paragraphs in Discussion overlapped to Introduction. I think the appropriate brief summary of the present results is preferable for readers at the beginning of Discussion.
Paragraph 1 has been deleted in the revised manuscript. We have retained paragraph 2 as we feel that it is important to remind the reader of the aims of the study